# Roll-AE: A Spatiotemporal Invariant Autoencoder for Neuronal Electro-physiology

## Abstract

Micro-electrode array (MEA) assays enable high-throughput recording of the electrophysiological activity in biological tissues, both in vivo and in vitro. While various classical and deep learning models have been developed for MEA signal analysis, the majority focus on in vivo experiments or specific downstream applications in vitro. Consequently, extracting relevant features from in vitro MEA recordings has remained largely dependent on particular curated features known as neural metrics. In this work, we introduce Roll-AE, a novel autoencoder designed to extract spatiotemporally invariant features from in vitro MEA recordings. Roll-AE serves as a foundational model that facilitates a wide range of downstream tasks. We demonstrate that 1) Roll-AE's embeddings outperform those from standard autoencoders across various classification tasks, and 2) Roll-AE's embeddings effectively characterize electrophysiological phenotypic traits in induced Pluripotent Stem Cells (iPSC)-derived neuronal cultures.

## 1 Introduction

In vitro micro-electrode array (MEA) assays provide a unique opportunity to obtain rich high-throughput electro-physiological data from induced Pluripotent Stem Cell (iPSC) (Takahashi & Yamanaka, 2006)-derived neuronal cultures (Fukushima et al., 2016; Kayama et al., 2018). MEA enables monitoring and recording of the extra-cellular action potentials in a non-invasive manner and provides valuable insights into the development and organization of neuronal networks (Novellino et al., 2011; Maeda et al., 2016). MEA has been particularly useful in the context of disease modeling, where excitability phenotypes are used to quantify the effects of genetic mutations and potential treatments of Parkinson's disease, Amyotrophic Lateral Sclerosis (ALS), Tuberous Sclerosis Complex (TSC), etc. (Woodard et al., 2014; Wainger et al., 2014; Winden et al., 2019)

Although MEAs on in-vitro cell cultures have increased in popularity over the past decade, existing methodologies to analyse this type of data remain limited. Most existing methods are based on a collection of hand-crafted features called neural metrics (Mossink et al., 2021; Wainger et al., 2014; Kim et al., 2020). These methods (Mack et al., 2014; Bryson et al., 2022; Kapucu et al., 2022; Passaro et al., 2021) apply principal component analysis, factor analysis, or similar dimension reduction techniques to distill and analyze electro-physiological recordings. Neural metrics generally include descriptive statistics on different activity patterns such as sporadic neuronal firing, synchronous firing across electrodes, rapid consecutive firings (bursts), and bursts across multiple electrodes (network bursts). Fig. 7a lists the neural metrics curated by the commonly used Axion Biosystems (Biosystems, 2024) algorithm. Analysis of these neural metrics comes with several limitations. First, data aggregation in a predefined manner results in a loss of resolution. A typical MEA recording involves action-potential data on 6-96 wells per plate and 8-64 electrodes per well at a millisecond sampling rate. This high-resolution data is then compressed to 30-40 predefined metrics, many of which are redundant because they are functionally linked to other metrics. This compression often leads to significant loss of information which adversely impacts the quality of both phenotypes and disease models. Second, these neural metrics depend on multiple manually picked hyperparameters (e.g., burst threshold). This may lead to high variance with respect to different experimental conditions (e.g., culture media, seeding cell density, etc.). Third, when an event either rarely occurs or does not occur in a recording (e.g., absence of bursts during low electro-physiological activity), a substantial portion of the neural metrics becomes unavailable. Handling such missingness in the data analysis stage is not trivial. Traditional imputation methods assume that the metric exists in reality but was

not observed. In this case, the metric is unavailable or undefined rather than not unobserved. This further creates a potential for neuronal metrics to be volatile and attain outlier values.

Over the past decade, deep learning models have revolutionized the fields of computer vision, natural language processing, video and speech recognition (LeCun et al., 2015; Chai et al., 2021; Chai & Li, 2019). Recent applications of deep learning in neuroscience have shown promising advancement especially in processing electroencephalogram (EEG) data or in vivo recordings (van Leeuwen et al., 2019; Schirrmeister et al., 2017; Buccino et al., 2018). Several recent statistical and machine-learning methods have been developed to understand the neuronal biology from live electro-physiological recordings of mouse-brains (Wu et al., 2017; Williams et al., 2020; Valente et al., 2022; Keeley et al., 2020). On passive MEA recordings of in vitro tissue cultures however, such applications are still limited. Recent works (Matsuda et al., 2022; Zhao et al., 2019), mainly deriving inspiration from computer vision, use of end-to-end convolutional neural networks (CNN) (Krizhevsky et al., 2012) on rasterized MEA signals for classifying gene knockouts and drug responses. Beyond models developed for specific tasks, foundational models such as autoencoders, or in general, feature extraction through self-supervised learning models, remains unexplored.

The MEA data modality is different from the image data modality and requires different calibration techniques. An important requirement for MEA is that the recordings remain invariant to shifts in time or changes in the orientation of the electrodes. When learning features from MEA recordings, it is important for a model to learn features that do not change if the recording is moved in time by a fixed amount, nor should they change if the spatial arrangement of the electrodes is permuted keeping the inter-electrode distances intact. These invariances are crucial as any temporal shift is determined by when the recording started and they have no biological relevance. Similarly, different spatial arrangements of the electrodes only reflect the arbitrary order in which they were arranged in the data matrix. Classical point process models (Snyder & Miller, 2012; Deutsch & Pfeifer, 1981) for analyzing spatiotemporal time-series data, in theory, can address this invariance issue. However, these models often require specific parametric assumptions, which can be overly restrictive for feature learning. Additionally, there is no widely adopted point process model for the analysis of in vitro MEA data; instead, in the broader field of neuronal electrophysiology, ad-hoc models (Amarasingham et al., 2006; Bogaard et al., 2009) are typically devised based on specific research hypotheses.

The spatiotemporal invariance for MEA recordings is analogous to orientational (rotation, translation, scale, mirror-flip) invariance of images for computer vision, where data augmentation is a standard technique of choice for model calibration (Perez & Wang, 2017). However, data augmentation does not guarantee that the encoded embeddings generated from an original image and a differently oriented image will be the same. Recent works (Burgess et al., 2024; Lohit & Trivedi, 2020) have proposed autoencoder architectures that guarantee such orientational invariance for 2D and spherical (3D) images by using orientation-equivariant convolution layers and a spatial pooling layer. Moreover, theoretical developments in the deep set literature (Zaheer et al., 2017) and group-equivariant methods (Cohen & Welling, 2016) provided necessary building blocks in understanding these invariance principles. Our work is deeply inspired by these recent developments.

In this paper, we show that augmentation techniques, even though slightly improve the model performances, still fail to learn important features for detecting subtle and complex firing patterns. For this reason, we propose Roll-AE, an novel autoencoder architecture that explicitly and completely calibrates for spatiotemporal invariance while extracting relevant features from MEA recordings. Roll-AE constructs invariant sets from given recordings and learns a set-to-set mapping with a low-dimensional bottleneck. Roll-AE is intended to be a foundational model for in vitro MEA recordings and its learned features can be used for multiple downstream tasks. We first demonstrate on a synthetic dataset that the Roll-AE embeddings have superior performance in identifying unique and complex firing patterns compared to standard autoencoders with augmentation. Then, we demonstrate multiple downstream applications of Roll-AE embeddings on an real iPSC-derived neuronal culture to illustrate that the proposed architecture captures meaningful multi-dimensional biological phenotypes useful for disease modeling and treatment discovery.

## 2 ROLL-AE

### 2.1 NOTATIONS

Let $x \in \Omega \subseteq \mathbb{R}^D$ be a $D$-length time-series (or *spike-train*) from a feature space $\Omega$ and let $x_i$ be the $i$-th element of $x$. Let $\mathbb{N}_D = \{0, 1, \ldots, D-1\}$, and $\pi_i : \Omega \to \Omega$ for $i \in \mathbb{N}_D$, be a cyclic permutation function where $\pi_i(x) = (x_{D-i+1}, x_{D-i+2}, \ldots, x_D, x_1, x_2, \ldots, x_{D-i})$. Intuitively, $\pi_i(x)$ cyclically shifts $x$'s elements by $i$ positions, and we can define such cyclic permutations as *shifts*. Let $\Pi(x) = \{\pi_i(x) : i \in \mathbb{N}_D\}$ be the set of all $D$ shifts of $x$ and $\Pi(\Omega) = \{\Pi(x) : x \in \Omega\}$ be the set of all $\Pi(x)$ where $x \in \Omega$.

### 2.2 STANDARD AUTOENCODER ARCHITECTURE

The standard autoencoder on the input feature space $\Omega$ is a map $f_{\theta,\phi} : \Omega \to \Omega$ with the encoder $g_\theta : \Omega \to \mathbb{R}^k$, decoder $h_\phi : \mathbb{R}^k \to \Omega$, and loss function $\ell : \Omega \times \Omega \to \mathbb{R}$. Here, $\theta$ and $\phi$ are parameters of the encoder and decoder, respectively. Let the output of $g_\theta(x)$ to be the encoded embedding of $x \in \Omega$ and $\mathbb{E} = \{g_\theta(x) : x \in \Omega\}$ to be the embedding space.

The goal of an autoencoder is to reconstruct a given input. Specifically, the network is trained to minimize a reconstruction loss $\ell(x, \hat{x})$ where $\hat{x} = f_{\theta,\phi}(x) = h_\phi(g_\theta(x))$ is the forward propagation reconstruction of a given input $x \in \Omega$. When $\Omega = \mathbb{R}^D$, a commonly adopted loss function is the mean-squared loss, defined as $\ell(x, \hat{x}) = \frac{1}{D} \sum_{i=1}^{D} (x_i - \hat{x}_i)^2$.

A limitation of the standard autoencoder is that it suffers from the lack of shift-invariance. This means that two spike-trains $x$ and $\pi_i(x)$, which are just shifted versions of each other, are interpreted as different spike-trains and encoded into different embeddings in the embedding space, i.e., $g_\theta(x) \neq g_\theta(\pi_i(x))$. In the electro-physiology context, a shift is purely determined by when the recording started and has no biological relevance. Hence, any difference between the embeddings of two shifted spike-trains should only represent noise or potentially confounding information.

Shift-invariance in time-series data is analogous to the rotational invariance of autoencoders applied in image processing, where we want the network to learn the same encoded embeddings from different rotations of the same image. A standard way to handle such invariances is to apply augmentations (Perez & Wang, 2017; Caron et al., 2021). In the context of MEA, this means, at each training iteration, a randomly selected shifted spike-train $\pi_i(x)$ is used as the input instead of $x$, and the reconstruction loss is calculated with respect to the original input $x$. Notice that even if augmentation encourages the network to reconstruct the original spike-train $x$, it does not guarantee that $g_\theta(x) = g_\theta(\pi_i(x))$ for all $i$, which means the invariance problem is only partially tackled by augmentation. Drawing inspiration from recently proposed methods (Burgess et al., 2024; Lohit & Trivedi, 2020) that achieves orientation-invariance for 2D and spherical (3D) images, in the following section we introduce a novel architecture that enforces shift-invariance directly in the encoding process, ensuring that $g_\theta(x) = g_\theta(\pi_i(x))$.

### 2.3 ROLL-AE

A *shift-invariant* loss $\rho : \Omega \to \Omega$ is defined such that the distance $\rho(\pi_i(x), \pi_j(x))$ between two shifts $\pi_i(x)$ and $\pi_j(x)$ of the same spike-train is zero for any $i, j \in \mathbb{N}_D$, or in other words, the shifted spike-trains are treated as the same spike-train. This can be achieved by defining the distance metric to be $\rho(x, x') = L(\Pi(x), \Pi(x'))$ where $L$ is a set-based loss such as Chamfer loss (Zhang et al., 2019), Linear assignment loss (Zhang et al., 2020), etc. Therefore, shift-invariance can be achieved in an autoencoder architecture by modifying the objective, specifically, by reconstructing entire sets $\Pi(x)$ instead of single spike-trains and back-propagating on the set-based loss function $L$. Since $L$ is invariant to the ordering of the elements in a set, it guarantees $\rho(\pi_i(x), \pi_j(x)) = L(\Pi(x), \Pi(x)) = 0$. Formally, Roll-AE can be defined as a network $f_{\theta,\phi} : \Pi(\Omega) \to \Pi(\Omega)$ with the encoder $g_\theta : \Pi(\Omega) \to \mathbb{R}^k$, decoder $h_\phi : \mathbb{R}^k \to \Pi(\Omega)$, and loss $L$ (see Eq. 1).

**Encoder** Since Roll-AE is a set-to-set mapping, the encoder is constructed based on ideas from the Deep Set literature (Zaheer et al., 2017; Soelch et al., 2019; Zhang et al., 2019). Explicitly, for any $\Pi(x) \in \Pi(\Omega)$, the encoder is defined as $g_\theta(\Pi(x)) = a(\{\widetilde{g_\theta}(x') : x' \in \Pi(x)\})$, where

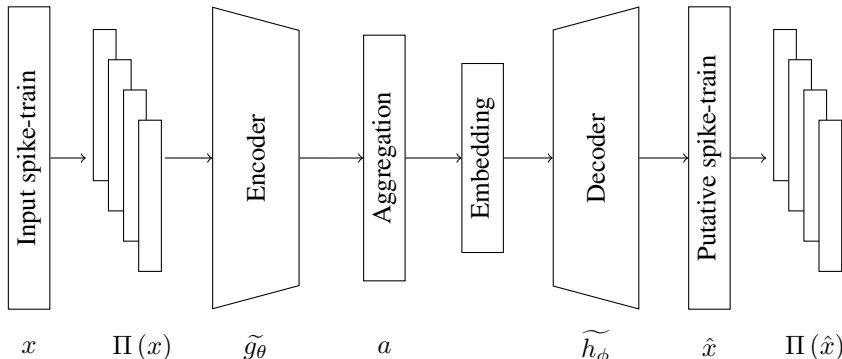

Figure 1: Roll-AE architecture. The input spike-train $x$ is converted into its cyclic permutation set $\Pi(x)$ which is passed to the encoder $\widetilde{g_\theta}$. The encoder outputs are grouped by the aggregation function $a$ into a single embedding. The decoder $\widetilde{h_\phi}$ reconstructs the putative spike-train $\hat{x}$ which is converted into its cyclic permutation set $\Pi(\hat{x})$. The reconstruction loss is computed between $\Pi(x)$ and $\Pi(\hat{x})$.

$\widetilde{g_\theta}(:) \Omega \to \mathbb{R}^k$ is a Multi-layer Perceptron (MLP) and $a$ is an aggregation function that aggregates the set of $k$-dimensional outputs from $\widetilde{g_\theta}$ into a single $k$-dimensional embedding. The purpose of the aggregation function is to make the learned embeddings invariant of the shifts. Notice that $a(\{\widetilde{g_\theta}(x') : x' \in \Pi(x)\}) = a(\{\widetilde{g_\theta}(x') : x' \in \Pi(\pi_i(x))\})$, and hence $g_\theta(\Pi(x)) = g_\theta(\Pi(\pi_i(x)))$. Typically, the average function is used as the aggregator, although other methods can be defined (Soelch et al., 2019). In the case of Roll-AE, the average function is utilized as the aggregation function.

**Decoder**  The decoder is defined as $h_\phi(e) = \Pi\left(\widetilde{h_\phi}(e)\right)$ for $e \in \mathbb{R}^k$, where $\widetilde{h_\phi} : \mathbb{R}^k \to \Omega$ is an MLP. A crucial challenge in constructing Deep Set autoencoders is finding a suitable mapping from the output of $\widetilde{h_\phi}$ to the space of sets. Roll-AE does not face this challenge as this mapping is deterministic and known. Therefore, the forward propagation of the overall Roll-AE architecture is defined as the mapping $f_{\theta,\phi}(\Pi(x)) = \Pi\left(\widetilde{h_\phi}(a(\{\widetilde{g_\theta}(x') : x' \in \Pi(x)\}))\right)$. Fig. 1 shows Roll-AE architecture. We note that the trainable parameters in Roll-AE are all contained in the encoder MLP $\widetilde{g_\theta}$ and the decoder MLP $\widetilde{h_\phi}$, which are functions that simply map $\Omega$ onto $\mathbb{R}^k$ and back. As a result, the number of trainable parameters in Roll-AE remains the same as a standard autoencoder.

**Reconstruction Loss**  Roll-AE uses the Linear Assignment loss as the reconstruction loss between $X = \Pi(x)$ and $X' = f_{\theta,\phi}(\Pi(x))$. Specifically, for any arbitrary ordering of the elements $X = \{x^{(1)}, x^{(2)}, \ldots, x^{(D)}\}$ and $X' = \{x'^{(1)}, x'^{(2)}, \ldots, x'^{(D)}\}$, and denoting $\Psi$ to be the set of all possible permutations (not just cyclic) of $(1, 2, \ldots, D)$, the Linear Assignment loss is defined as,

$$L(X, X') = \min_{\psi \in \Psi} \sum_{i \in \mathbb{N}_D} \ell\left(x^{(i)}, x'^{\psi(i)}\right). \tag{1}$$

Computing the Linear assignment loss, in general, is extremely expensive with complexity $O(s^3)$ using the Hungarian algorithm, where $s$ is the cardinality of $X$ (in this context, $s = D$). However, in our problem, since both the sets $X$ and $X'$ are closed under the cyclic permutation operation, the computation can be substantially improved to $O(r)$ complexity where $r$ is the order of the cyclic permutation operation (in this context, $r = s = D$). Lemma A.1 then simplifies this reconstruction loss as $L(\Pi(x), \Pi(\hat{x})) = D\left[\min_{i \in \mathbb{N}_D} \ell(\pi_i(x), \hat{x})\right]$, where $\hat{x} = \widetilde{h_\phi}(a(\{\widetilde{g_\theta}(x') : x' \in \Pi(x)\}))$ is the putative output train. Further, it trivially follows from Lemma A.1 that the above expression also applies to Chamfer loss.

**Stochastic Shift-invariance**  Roll-AE further implements stochastic shift-invariance, i.e., instead of passing all possible shifted spike-trains in $\Pi(x)$ on each forward propagation iteration, it uniformly samples spike-trains from $\Pi(x)$ with sampling rate $\tau$ and uses the set of sampled spike-trains as

inputs. Once the model is trained, the final embeddings can then be calculated by running a final forward propagation of the encoder with the entire $\Pi(x)$ as input. This ensures the invariance in the final embeddings remains intact. In Appendix C, we demonstrate through extensive simulation studies that stochastic shift-invariance can perform as good or sometimes even better than complete shift-invariance ($\tau = 1$) while reducing the memory requirement and computation time.

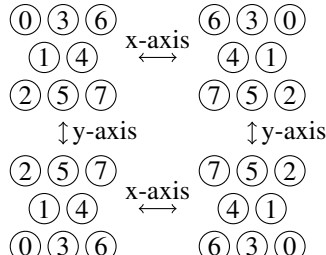

(a) A well and electrodes used to record electrophysiological activity.

(b) All possible mirror symmetries of electrodes in a well.

Figure 2: Electro-physiological activity recording and electrode symmetry analysis in a well. On the left, an image of an well and electrodes from Axon Biosystems (Biosystems, 2024). The eight circular dots near the center are the recording electrodes, and are arranged in this 3-2-3 (row-wise) arrangement; on the right; all possible electrode symmetries and corresponding indices. In a typical recording from (Biosystems, 2024), the electrodes are labeled following the pattern on the top left corner.

**Generic Permutations**  Roll-AE is not limited to temporal shifts or permutation of input spike-trains. The above definitions and properties hold as long as $\Pi(x)$ is a closed group with respect to any class of cyclic permutations. This allows Roll-AE to handle other interesting classes of permutation that are relevant for our biological assays. For instance, mirror symmetries can be used to permute electrodes in a well and enforce spatial invariance on our embeddings. When multiple spike-trains are simultaneously recorded by different electrodes placed on the same cell culture, they need to be arranged and inputted into the model in a specific reference order. However, this order is arbitrary, and any permutation should be biologically equivalent to the reference order as long as the inter-electrode distances are preserved. Such permutations are given by the mirror flip operation. Fig. 2a shows an image of an actual eight-electrode well used to record neuronal electrophysilogical activity using an Axion Biosystems (Biosystems, 2024) MEA machine. Fig. 2b shows all possible electrode orderings with the top-left graph matching the reference orientation. Notice that the four permutations defined by mirror symmetry form a closed group under the mirror flip operation which further is a special case of cyclic permutations. Folding in the mirror symmetries into the previously defined Roll-AE architecture results in the cardinalities of the input and output sets to become $4D$ with $D$ temporal permutations, and four spatial permutations.

In Sec. 3.2, we will consider an actual dataset collected using the system depicted in Fig. 2a and train Roll-AE using both temporal shift and electrode mirror-symmetry invariances.

## 3  EVALUATIONS

In this section we evaluate the proposed architecture on two case studies. First, we compare a standard autoencoder, a standard autoencoder with augmentation, and Roll-AE on a synthetic dataset. Second, we train and evaluate Roll-AE on a dataset of electro-physiology recordings of neuronal induced Pluripotent Stem Cells (iPSC) (Takahashi & Yamanaka, 2006) subject to different Small Interfering RNA (siRNA) (Fire et al., 1998) treatments.

## 3.1 SYNTHETIC DATA EVALUATION

Just like in real recordings, we simulate recordings with spike-trains on eight electrodes. To mimic the electro-physiology and firing patterns of real neuronal cultures, our synthetic dataset has four tunable parameters that determine the level of activity along four different types of source-events, namely sporadic spikes (spike), sporadic single-channel bursts (burst), cyclic bursts (cycle), and network bursts (network). These parameters are formally defined as follows: *spike*: the probability of a sporadic firing event at a particular time instance; *burst*: the probability of a sequence of firing events starting at a particular time instance for a random duration; *cycle*: presence or absence of a repeating pattern of sequence of firing events with a random phase and duration; *network*: presence or absence of network bursts where sequences of firing events are recorded simultaneously across multiple electrodes. Each network burst starts with an originating electrode, and the probability of observing firing events in other electrodes depend on their proximity to the originating electrode.

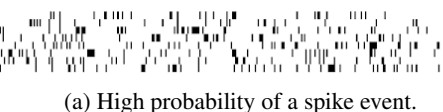

(a) High probability of a spike event.

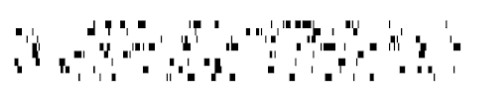

(b) High probability of a burst event.

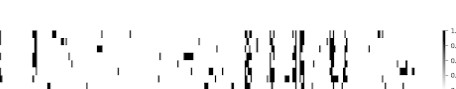

(c) Presence of cyclic bursts.

(d) Presence of network bursts.

Figure 3: Examples of synthetically generated spike-trains from eight electrodes tuning the probabilities of spike, burst, cycle, or network firing events.

Fig. 3 depicts some examples of firing patterns where one type of firing event was kept at a high probability (or present) and the others are kept at a low probability (or absent). For instance, Fig. 3a shows a recording obtained with high spike but low burst probabilities with no cycle or network behaviors, resulting in a dense randomly scattered stack of spike-trains. Similarly, Fig. 3d depicts a recording obtained with the presence of network behavior, but with low spike and burst probabilities, and absence of cyclic bursts. This results in sparse but vertically aligned spike-trains, mimicking a low sporadic activity but a well-synchronized network. The explicit details of the data generating model for the synthetic data are presented in Appendix B. For each of the tunable firing parameters, we considered two classes, high probability and low probability classes for the spike and burst parameters, and present and absent classes for cyclic and network parameters. This resulted in a total of 16 combined classes of firing patterns. For each of these 16 classes, we generated 500 synthetic recordings, each recording having 300-length spike-trains across eight channels (electrodes).

**Model Training** We trained three models: a standard autoencoder, a standard autoencoder with augmentation, and Roll-AE on the synthetic dataset. The hyperparameters (such as training batch-size, embedding dimension etc.) for each model were selected based on a two-step training/validation scheme (see Appendix D for training details). The embeddings generated by the trained models were then compared based on downstream classification tasks.

We trained five classifiers (with $70/30\%$ training/validation data split) using the embeddings from each autoencoder: four binary classifiers (e.g., high vs low probability of spikes, high vs low probability of a bursts, etc.) and a multi-class classifier with 16 classes encompassing all combinations of our synthetic dataset parameters (e.g., low spike, low burst, no cycle, no network vs low spike, low burst, no cycle, present network vs low spike, low burst, present cycle, present network, and so on). Specifically, we trained logistic regressors with L2 regularization with the penalty parameter determined using a 4-fold cross-validation. The best trained logistic regressor was then used to make predictions on the validation data.

**Results** Fig. 4 reports the obtained accuracies together with the confusion matrices for the multi-class task. Overall, Roll-AE outperforms the standard autoencoders across all classification tasks (see Fig. 4a). Remarkably, Roll-AE achieved a +30% accuracy on the hardest multi-class task. It is also worth noticing, that augmentation improves the standard autoecoder across all tasks (except

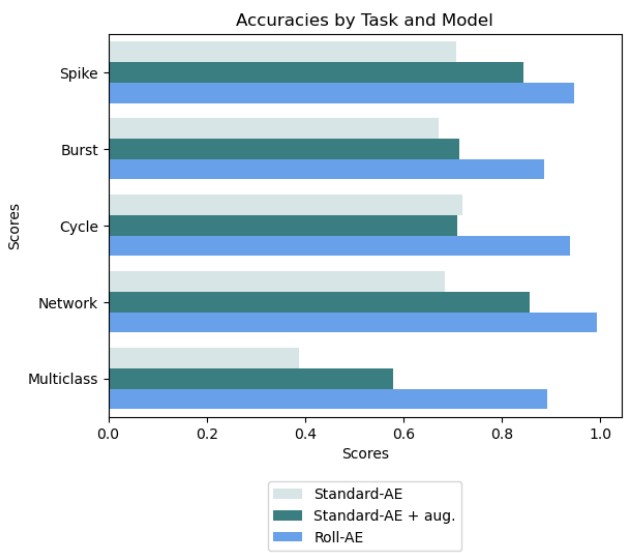

(a) Accuracy comparison between autoencoders across tasks. Roll-AE achieves highest accuracy across all binary and mutliclass tasks.

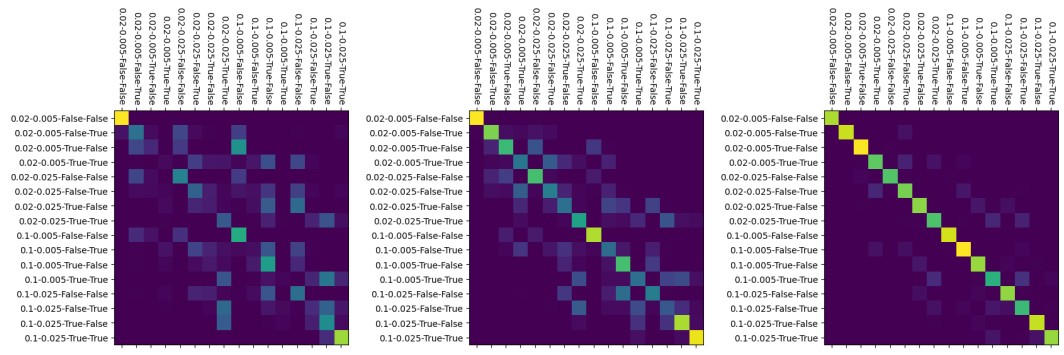

(b) Confusion matrices of standard (left), standard with augmentation (center), and Roll-AE (right) autoencoders for multi-class classification. The 16 class labels were each formatted as {spike probability}-{burst probability}-{cycle present}-{network present}.

Figure 4: Roll-AE consistently outperforms standard autoencoders (both with and without augmentation) on all binary and multiclass classification tasks.

*cycle*), while still being sensibly inferior to Roll-AE. Fig. 4b reports the confusion matrices of the three models for the multiclass task. These plots highlight how Roll-AE embeddings lead to a high predictive accuracy in our downstream classification tasks while the two standard autoencoders tend to misclassify similar classes in a multi-class regime.

## 3.2 SIRNA TREATMENT EVALUATION

As a second case study, we applied the three models on a real electrophysiological dataset obtained from induced Pluripotent Stem Cells (iPSC) (Takahashi & Yamanaka, 2006) derived neurons subject to Small Interfering RNA (siRNA) (Fire et al., 1998) treatments. Technically, iPSCs are cells that have been reprogrammed from skin or blood cells to become other types of cells, in our case, neurons. siRNAs are artificially synthesized RNA molecules commonly used in molecular biology for silencing genes of interest. In our case, we apply a double siRNA treatment: the first to silence a gene to trigger the disease, the second to investigate a possible cure to counteract the effect of the first intervention.

Our neuronal culture was organized on two 96-well plates. Half of the sample set was subject to a siRNA knockdown (siKD) of the gene of interest, mimicking the effect of the considered disease, while the other half was subject to a non-targeting control siRNA sequence (NTS) designed to target no known genes. The NTS treatment is our negative control, i.e., a condition that does not affect the neuron state. Both siKD and NTS samples have been then treated with 24 different siRNA potential cures. Among these, there is an additional negative control NTS and a positive control CTRL+ known to reduce neuronal excitability. For each condition, we cultured 4 replicates recorded at nine different days up to 24 days in vitro. In total we obtained 2 (siKD or NTS) $\times$ 24 (siRNA treatments) $\times$ 4 (replicates) $\times$ 9 (days in vitro) $\times$ 8 (electrodes) = 13824 raw spike-trains. Each spike-train was recorded with one milli-second sampling rate for 10 minutes. We bucketized the raw spike-trains to 500 milli-second bins to end up with spike-train length of $D = 1200$. To train the models, we first performed a hyperparameter selection of training batch-size, embedding dimension, learning rate, and shift-sampling rate (see details in Appendix D). Models with the selected hyperparameters were then trained to generate the final embeddings. First, we compare the embeddings on an siKD phenotyping task.

**Results - siKD Phenotyping**   We compared the embeddings from the three models based on their performance in the classification of siKD wells from NTS wells on each of the 24 siRNA treatments. Biologically, the effect of siKD is very subtle, and discovering a classifier with good classification accuracy helps uncover subtle phenotypes in our disease model. We applied a leave-one-well-out approach, where at each iteration, we left out all the recordings from a particular well with a particular siRNA treatment, trained a siKD vs NTS classifier on the rest of the wells, and predicted the recordings of the left-out well. Logistic regressors with L2 regularization and 4-fold cross-validation were used as the classifiers.

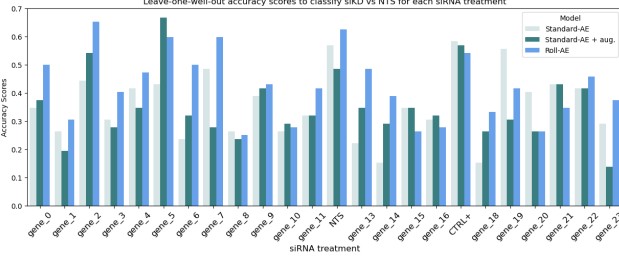

Figure 5: Accuracy comparison between autoencoders for classification of siKD vs NTS on different siRNA treatments. Roll-AE has the highest accuracy for 15 our of the 24 treatments, in particular, on NTS.

The average classification accuracies are presented in Fig. 5. For 15 out of the 24 treatments, Roll-AE embeddings had the highest classification accuracy, and most importantly, Roll-AE had the highest accuracy on NTS. Next, we demonstrate the application of the Roll-AE embeddings on two downstream tasks: treatment clustering and neural metrics credentialing.

**Results - Treatment Clustering** For this study, we used Roll-AE embeddings to extract biological insights and characterize treatment similarities. To do so, we considered cells cultured for 24 days in vitro, reduced the dimension of our embeddings to 10 principal components explaining at least 95% of the variance, computed the centroids of each treatment cluster, and calculated the pairwise distances between treatment centroids. The obtained results for NTS and siKD treated cells are organized in the two dendrograms shown in Fig. 6a and Fig. 6b, respectively.

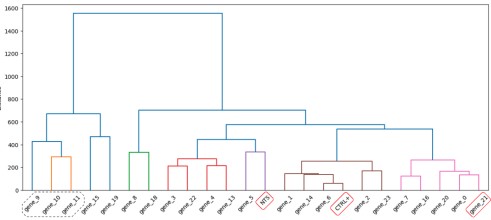
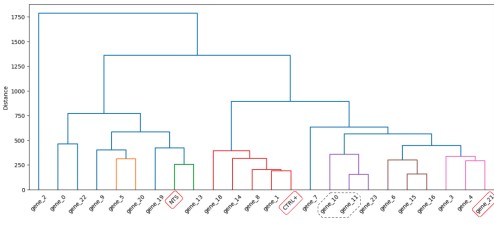

(a) Pairwise distances between genes for NTS cells.  (b) Pairwise distances between genes for siKD cells.

Figure 6: Dendrograms of embedding pairwise distances between gene centroids applied to unperturbed (NTS) and diseased (siKD) cells.

For cells with NTS, we observe a clear distinction between CTRL+ and NTS treatments, indicating that the model effectively differentiates CTRL+ (known to reduce cell excitability) from unperturbed NTS cells. On the other hand, gene_21, which clusters far from CTRL+, is known for inducing hyperexcitability and stimulating neuronal activity. Furthermore, gene_9, gene_10, and gene_11, which clustered together, are all components of the same signaling pathway involved in stress and inflammatory responses.

For cells with siKD, we observe again that NTS and CTRL+ treatments form distinct clusters, with gene_21 clustering on the opposite side of the hyperexcitability spectrum of the dendrogram. Interestingly, gene_10 and 11 are still clustering together but gene_9 does not. Additionally, gene_3 and gene_4, which were grouped as hyperexcitable in the NTS cells study, now cluster among hypoexcitable treatments. This suggests that these genes interact differently with the induced disease state, leading to different types of neuronal activity depending on the disease state of the cell.

This study is an example of how extracting insights and clustering gene treatments can aid biologists in formulating hypotheses and identifying recurring patterns across various gene treatments.

**Results - Neural Metrics Credentialing** The goal of this study is to verify whether the obtained embeddings retain relevant information on the curated neural metrics provided by the Axon Biosystems instrument that are explicitly computed from raw recordings. The full metric list is reported in Fig. 7a.

To asses the ability of our embeddings to retain relevant information on these curated neural metrics, for each metric, we trained a Ridge regressor with an $L2$ penalty of 1.0. This was trained on 80% of the entire dataset and then used to predict each metric on the remaining 20%. The correlation (r-score) between the predicted and observed neuronal metrics from the validation data is shown in Fig. 7a. The metrics are ranked from the highest to the lowest correlation.

Roll-AE embeddings predict most metrics with high correlation. Out of 44 regressed metrics, 25 had r-score above 0.75, 9 between 0.25 and 0.75, and 10 between -0.1 and 0.25. Fig. 7b illustrates some scatter plots of actual metric values against the predicted ones. We observed that metrics useful for phenotypic analyses, including firing, spike, or burst counts and rates, are accurately captured and predicted. However, certain metrics, such as those related to the inter-burst interval (IBI coefficient) (Di Credico et al., 2021), demonstrated lower correlations. We hypothesize that this discrepancy may be attributed to the selected binning size during the compression of the raw signal, which can potentially eliminate inter-burst information.

Overall, this study demonstrates Roll-AE's embedding effectiveness in capturing explicit neural metrics and hence their potential as a tool in phenotypic analyses and downstream tasks.

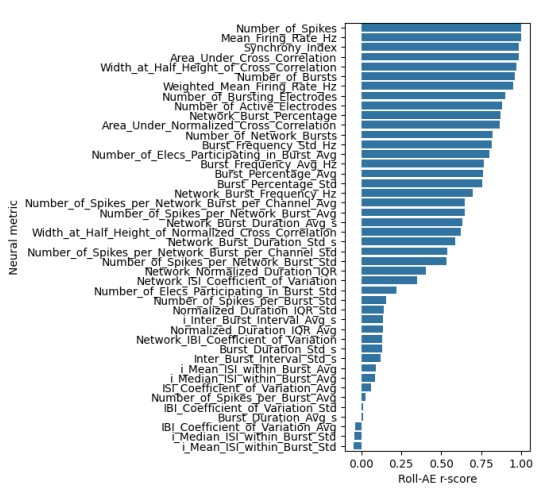 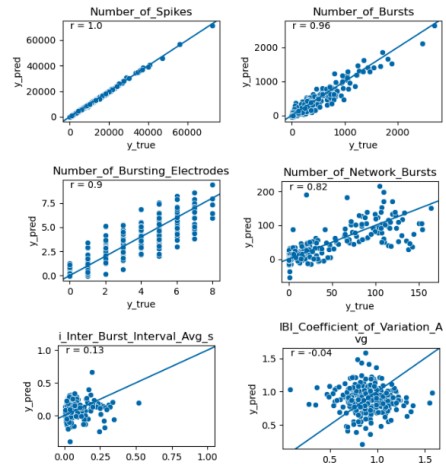

(a) r-scores between observed and predicted neural metrics from Roll-AE embeddings. High r-scores indicate Roll-AE's embedding ability to capture explicit neural metrics.

(b) Examples of observed vs predicted neural metrics metrics from Roll-AE embeddings with high (top row) and low (bottom row) r-scores.

Figure 7: Credentialing the Roll-AE embeddings by evaluating its performance in predicting the neural metrics.

# 4 CONCLUSION

In this paper, we proposed Roll-AE, a novel spatiotemporal invariant autoencoder for feature extraction from passive recordings of MEA assays. By leveraging and explicitly imposing the invariances in the architecture, Roll-AE can extract features that are relevant for identifying unique firing patterns. On synthetic data, we demonstrated that the Roll-AE embeddings far outperform standard autoencoders (with or without augmentation) in discriminating different individual source-events. Roll-AE is particularly accurate in the multi-class classification task and showed a $+30\%$ accuracy gain compared to standard autoencoders suggesting Roll-AE's ability to identify features relevant to complex and subtle phenotypes. On the siRNA experiment data, we further considered multi-faceted downstream applications of Roll-AE generated embeddings. We demonstrated the superior performance of Roll-AE embeddings in discriminating siKD from NTS highlighting its use in phenotype discovery. The concordance of the relative clustering of treatments with previous biological evidences supports the validity of these machine learnt features. Finally, we showed that these embeddings retain explicit metrics and can be used to predict manually curated features.

The original formulation of Roll-AE came with a few limitations, mainly in regard to computational efficiency. Constructing the entire set of cyclic permutations and evaluating the Linear Assignment loss can be computationally demanding. To tackle this issue, we have proposed the stochastic shift-invariance approach. Another possible approach is to apply discrete time Fourier transformation to each spike-train to transform them from time domain to frequency domain and apply invariance to analogous operations to temporal shifts. In our siRNA experiment example, we found that the embeddings were not able to predict some neural metrics well, predominantly those with inter-spike interval-related metrics. This could be a consequence of the adopted 500 milli-seconds bin size. While we have demonstrated its efficacy using in vitro MEA data, the autoencoder design could be easily generalized for other types of data where such spatiotemporal invariance is relevant. Other potential use cases for such architectures would be to identify arrhythmia from ECG data and anomaly detection from sensors.

In conclusion, Roll-AE provides a foundational model for extracting features from in vitro MEA recordings. The features from Roll-AE enables better identification of unique electro-physiological activity patterns from MEA recordings, and can be used for a multitude of downstream applications including the identification of complex cellular phenotypes of different treatments such as siRNA knock-down, gene knock-outs etc.

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

## A  MATHEMATICAL RESULTS

**Lemma A.1.** *For two sets $\Pi(x)$ and $\Pi(y)$; $x, y \in \Omega$,*

$$\min_{\psi \in \Psi} \sum_{i \in \mathbb{N}_D} \ell\left(\pi_i(x), \pi_{\psi(i)}(y)\right) = D \times \left[\min_{i \in \mathbb{N}_D} \ell\left(\pi_i(x)\right), y\right].$$

*Proof.* First, we note that, for any $i \in \mathbb{N}_D, \psi \in \Psi$,

$$\ell\left(\pi_i(x), \pi_{\psi(i)}(y)\right) \geq \min_{j,j' \in \mathbb{N}_D} \ell\left(\pi_j(x), \pi_{j'}(y)\right),$$

which implies

$$\min_{\psi \in \Psi} \sum_{i \in \mathbb{N}_D} \ell\left(\pi_i(x), \pi_{\psi(i)}(y)\right) \geq D \times \left[\min_{j,j' \in \mathbb{N}_D} \ell\left(\pi_j(x), \pi_{j'}(y)\right)\right]. \tag{2}$$

Lets denote $(j^*, j'^*) = \arg\min_{j,j'} \ell\left(\pi_j(x), \pi_{j'}(y)\right)$. Further, let the permutation $\psi* \in \Psi$ to be a cyclic permutation that maps $j^*$ to $j'^*$, explicitly, $\psi^*(j^*) = j'^*$. Then,

$$D \times \left[\min_{j,j' \in \mathbb{N}_D} \ell\left(\pi_j(x), \pi_{j'}(y)\right)\right] = D\ell\left(\pi_{j^*}(x), \pi_{j'^*}(y)\right)$$

$$= D\ell\left(\pi_{j^*}(x), \pi_{\psi^*(j^*)}(y)\right) = D\ell\left(\pi_{i^*}(x), \pi_{\psi^*(i^*)}(y)\right).$$

The above is true for any $i^* \in \mathbb{N}_D$. Therefore, for $\psi = \psi^*$, the equality holds in 2, i.e.,

$$\min_{\psi \in \Psi} \sum_{i \in \mathbb{N}_D} \ell\left(\pi_i(x), \pi_{\psi(i)}(y)\right) = D\ell\left(\pi_{i^*}(x), \pi_{\psi^*(i^*)}(y)\right) \quad \forall i^* \in \mathbb{N}_D.$$

The final part of the proof holds by selecting $i^*$ such that $\psi^*(i^*) = 0$,

$$\ell\left(\pi_{i^*}(x), \pi_0(y)\right) = \min_{i \in \mathbb{N}_D} \ell\left(\pi_i(x), y\right).$$

$\square$

## B  SYNTHETIC DATA GENERATION

The synthetic data are simulated as normalized binary tensors. Lets denote the synthetic data corresponding to the tunable parameters $\beta_s, \beta_b, \beta_c, \beta_n$ as $X(\beta_s, \beta_b, \beta_c, \beta_n)$ with dimensions $N \times E \times D$, where $N = 500$ is the number of recordings, $E = 8$ is the number of electrodes, and $D = 300$ is the recording time duration. The parameter $\beta_s = \{0.02, 0.1\}$ represents the probability of a sporadic firing event. The parameter $\beta_b = \{0.005, 0.025\}$ represents the probability of a sporadic sequence of multiple firing event, or a burst on a single electrode. The parameter $\beta_c = \{0, 1\}$ represents the absence or presence of cyclic burst firing pattern, and the parameter $\beta_n = \{0, 1\}$ represents the absence or presence of network burst firing pattern. Let the indices $n$, $e$, and $d$ represent single instances of recordings, electrodes, and timepoints. Let us also denote the four different firing patterns sporadic single firing, sporadic single-channel burst, cyclic single-channel burst, and network burst as *source events*.

To simulate the synthetic recordings, first, four source-specific binary recordings were simulated corresponding to each of the four different source events, and then those recordings were combined using the binary OR ($\vee$) operation. This means, at any given time-point on a given electrode, a neuronal firing can be observed due to any combination of the source events. The synthetic data simulation algorithm is as follows:

**Algorithm B.1** (Algorithm to simulate the synthetic recordings)**.** *First, initialize the following parameters: Firing frequency within a burst $\gamma_b = 0.9$, probability of a network burst starting at a given time point $\gamma_n = 0.035$, and network decay factor $\delta = 0.8$.*

1. ***Sporadic single firings (Spike)****: Simulate* $Z^{(S)}_{n,e,d} \sim Bernoulli(\beta_s)$ *i.i.d.*

2. ***Sporadic single-channel bursts (Burst)****:*

   (a) *Initialize* $Z^{(B)} = 0$.
   (b) *Simulate burst initiation indicators* $S_{n,e,d} \sim Bernoulli(\beta_b)$ *i.i.d.*
   (c) *For each* $S_{n,e,d} = 1$,
      i. *Select a burst duration* $\Delta = min(D, \Delta^*)$ *where* $\Delta^* \sim DiscUnif(\{3,4,5\})$, *the discrete Uniform distribution.*
      ii. *Simulate the source-specific firings* $Z^{(B)}_{n,e,i} \sim Bernoulli(\gamma_b)$ *for* $i = d, \dots, d + \Delta$.

3. ***Cyclic single-channel bursts (Cycle)****:*

   (a) *Initialize* $Z^{(C)} = 0$. *If* $\beta_c = 0$, *then skip to step 4. Else, move to the next step.*
   (b) *For each recording* $n$ *and electrode* $e$,
      i. *Select a cycle period* $Q_c \sim DiscUnif(\{15, 16, \dots, 19\})$ *and phase* $P_c \sim DiscUnif(\{0, 1, \dots, 14\})$.
      ii. *Set the burst initiation indicators* $S_{n,e,d} = 1$ *if* $(d - P_c)$ *is divisible by* $Q_c$, $0$ *otherwise.*
      iii. *For each* $S_{n,e,d} = 1$,
         A. *Select a burst duration* $\Delta = min(D, \Delta^*)$ *where* $\Delta^* \sim DiscUnif(\{3,4,5\})$, *the discrete Uniform distribution.*
         B. *Simulate the source-specific firings* $Z^{(C)}_{n,e,i} \sim Bernoulli(\gamma_b)$ *for* $i = d, \dots, d + \Delta$.

4. ***Network bursts (Network)****:*

   (a) *If* $\beta_n = 0$, *set* $Z^{(N)} = 0$ *and skip to step 5. Else, move to the next step.*
   (b) *Simulate burst initiation indicators* $S_{n,e,d} \sim Bernoulli(\gamma_n/E)$ *i.i.d.*
   (c) *For each* $S_{n,e,d} = 1$,
      i. *Denote* $e$ *to be the starting electrode, and* $d$ *to be starting time-point for the network burst.*
      ii. *Select a burst duration* $\Delta = min(D, \Delta^*)$ *where* $\Delta^* \sim DiscUnif(\{3,4,5\})$, *the discrete Uniform distribution.*
      iii. *Simulate the source-specific firings* $Z^{(N)}_{n,e',i} \sim Bernoulli\left(\gamma_b \delta^{\alpha(e,e')}\right)$ *for* $i = d, \dots, d + \Delta$ *and* $e' \in \{1, \dots, E\}$. *Here,* $\alpha(e, e')$ *represents the physical distance between the electrodes* $e$ *and* $e'$ *assuming the distance between electrodes* $0$ *and* $1$ *in the configuration described in Fig. 2 to be one unit.*

5. ***Combine the recordings****: Obtain the combined recording* $Z = Z^{(S)} \vee Z^{(B)} \vee Z^{(C)} \vee Z^{(N)}$.

6. ***Normalize****: The final dataset* $X(\beta_s, \beta_b, \beta_c, \beta_n)$ *is obtained by normalizing the dataset* $Z$.

## C  EVALUATING STOCHASTIC SHIFT-INVARIANCE

Here, we evaluate the accuracy, computation time, and memory requirements of the Roll-AE model under the stochastic shift-invariance strategy based on the synthetic data (generative model B). We trained the Roll-AE model with different training batch-sizes, embedding dimensions ($k$), and shift-sampling rates ($\tau$). We evaluated the accuracy under each selection of hyperparameters using the same evaluation scheme outlined in D. All models were trained with learning rate 0.0001.

Fig. 8 shows the classification accuracy, computation time, and memory requirements for each selection of the hyperparameters. The accuracies were similar across all choices of hyperparameters, and except for the case with batch-size $= 64$ and embedding dimension $k = 128$, the shift-sampling rate $\tau = 0.01$ resulted in the best accuracy for the 16-class classification task on the second-level validation dataset (see D). As expected, the computation time was the longest for $\tau = 1$ which implies the entire set $\Pi(x)$ was used in each pass of model training. However, the strongest contributor to

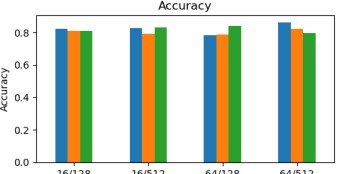 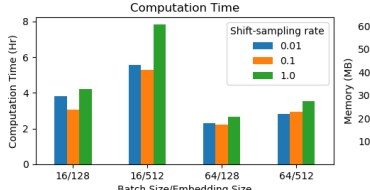 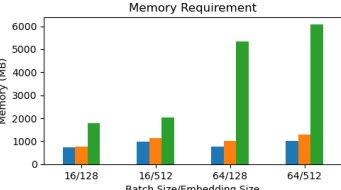

Figure 8: Accuracy, Computation time (Hour), and Memory requirement (MB) for different training batch-sizes and embedding dimensions ($k$), with respect to different shift-sampling rates ($\tau$). All models were trained on an NVIDIA® V100 GPU.

the computation time was batch size, with batch-size 16 requiring $\sim 1.5 - 2$ times more computation time than batch-size 64.

The most important benefit of the stochastic shift-invariance is in the memory requirement. Training with $\tau = 0.01$ required half as much memory for batch-size = 16, and nearly one-seventh as much memory for batch-size 64 compared to $\tau = 1$. By requiring substantially lower memory than training with complete shift-invariance (entire $\Pi(x)$), in large datasets, the stochastic shift-invariance strategy can allow for training with larger batch-sizes on limited GPU memory, which in turn can help reduce the computation time without impacting the accuracy.

## D  MODEL TRAINING AND HYPERPARAMETER SELECTION

### D.1  MODEL TRAINING

For the synthetic data, we trained the three models with different hyperparameters (listed in Table 1) on 70% of the data (training) randomly selected, and then applied the trained model on the remaining 30% of the data (validation) to obtain the embeddings. The embeddings of the validation dataset were then further split into a second-level of 70%-30% training/validation data to evaluate the predictive accuracy of those embeddings for the 16-class classification task. For this purpose, a 16-class logistic regression classifier with L2-regularization was trained on the second-level training data, and then the classes were predicted on the second-level validation data. The penalty parameter for the L2-regularization was selected based on four-fold cross-validation. Using the two levels of training/validation data splits, we ensured that both the autoencoder model, and the downstream classifier are generalizable to previously unseen data. For each of the three models, whichever hyperparameters led to the highest predictive accuracy in the second-level validation data, were selected and the models with those selected hyperparameters were then trained on the entire dataset to generate the final embeddings.

Similar two-level validation approach was taken for selecting hyperparameters in the siRNA data. We first trained the three models with the same set of hyperparameters (listed in Table 1) and generated the embeddings based on a 70%-30% training/validation data split. Then, with the embeddings of the validation dataset, we evaluated a logistic regression classifier (with L2 regularization) of the siKD vs NTS samples based on a second-level 70% − 30% split of the data. The hyperparameters which led to the highest accuracy in the second-level validation data were selected for each model. Finally, we applied the three models with the selected hyperparameters on the entire dataset to generate the final embeddings.

### D.2  HYPERPARAMETERS

Table 1 lists all choices of the training parameters that were considered to train the three autoencoder models. For the standard autoencoder with augmentation, two augmentation sampling schemes were evaluated, namely Uniform and Half-mass. For the Uniform sampling scheme, on each epoch, the augmented recording was randomly selected from the set of all possible shifted (and mirror-flipped) recordings uniformly. On the contrary, for the Half-mass sampling scheme, with probability 0.5 the original recording was used as the augmented spike-train, and the with the rest 0.5 probability, the other shifted (and mirror-flipped) recordings were uniformly selected. The best training parameter

choice for each autoencoder model was determined based on the performance of the embeddings in the multi-class classification task for synthetic data, or in the binary classification of siKD and NTS in the siRNA experiment data. These parameter choices are highlighted in Table 1 in bold.

Table 1: Choice of training parameters for the three autoencoder models. The best choice of parameters for each autoencoder model are highlighted in bold font (for the synthetic data) or with an asterisk (for the siRNA experiment data).

| Training parameter | Standard AE | Standard AE + aug. | Roll-AE |
|---|---|---|---|
| Training batch-size | **8**, 16, 32, 64* | 8, **16**\*, 32, 64 | 8, 16, 32, **64**\* |
| Embedding dimension ($k$) | 128*, **256**, 512 | 128, **256**\*, 512 | 128, 256, **512**\* |
| Learning rate | 0.0001, **0.001**\*, 0.01, 0.1 | **0.0001**\*, 0.001, 0.01, 0.1 | **0.0001**\*, 0.001, 0.01, 0.1 |
| Augmentation scheme | N/A | **Uniform**, Half-mass* | N/A |
| Shift-sampling rate ($\tau$) | N/A | N/A | **0.01**\*, 0.05, 0.1, 1 |

All models had two hidden layers in each of their encoder and decoder MLPs. The hidden layers in the encoder MLPs had $4k$ and $2k$ neurons sequentially, where $k$ is the embedding dimension. Conversely, the hidden layers in the decoder MLPs had $2k$ and $4k$ neurons sequentially. Counting the $k$ parameters for the output layer of the encoder MLP and the $ED$ parameters for the output layer of the decoder MLP, our models had a total of $(13k + ED)$ trainable parameters. Mean-squared error loss was used for the standard autoencoders (with or without augmentation), and Linear assignment loss was used for Roll-AE. Adam optimizer Kingma & Ba (2017) was used for the back-propagation in all models. Each model was trained for 200 epochs.

