# OpenReview forum: "Roll-AE: A Spatiotemporal Invariant Autoencoder for Neuronal Electro-Physiology"
_ICLR.cc/2025/Conference — Submitted to ICLR 2025_

### Official Review · Reviewer_R3Ce · 2024-10-18

**Soundness:** 1
**Presentation:** 2
**Contribution:** 1
**Rating:** 3
**Confidence:** 4

**Summary:**

The authors develop an autoencoder model to analyze the firing patterns of neurons recorded in vitro, to overcome the shortcomings of using standard neural metrics (e.g. firing rates, synchrony, bursts). They compare their model to a standard autoencoder and an autoencoder trained using data augmentation on a synthetic dataset and electrophysiological recordings.

**Strengths:**

The proposed autoencoder model seems novel and the goal of extracting meaningful features from neural recordings is important. The model shows superior performance compared to the other autoencoder models on predicting the neural measures on the synthetic dataset. Furthermore, the clustering of the genes in Fig. 6 is interesting and may prove useful in the development of new therapies.

**Weaknesses:**

Although the paper is interesting, I have noted several shortcomings that question whether the author's intended goal of improving upon standard neural measures is achieved.

1. **Stated weaknesses of neural measures are not addressed**
In the introduction the authors state that neural metrics depend on manually picked hyperparameters, but I don't see how the autoencoder fixes this problem as its embeddings are also dependent on particular hyperparameters (e.g. number of units). I also did not find any experiments exploring the effect of hyperparameters on the autoencoder model. The authors state the autoencoder is a foundation model. Does this mean it can be trained once and used on different datasets without re-training from scratch? If so, this should be explored in the paper.

2.  **Unclear how the autoencoder is better than neural measures** The authors explore predicting neural measures from the embeddings in the autoencoder on the synthetic and electrophysiological datasets. However, this does not tell us much regarding the interpretability of the autoencoder embeddings (or how they are better than neural measures). The authors employ PCA to analyze the autoencoder embeddings in Fig. 6 to explore the relationship between genes. However, it is unclear if the same relationships could not be found when employing PCA on the neural measures. Besides, the authors state that the use of PCA and related methods (line 37-40) is a shortcoming of conventional MEA analysis, but then proceed to use it in their own analysis. It is unclear what is gained by using the autoencoder.

2.  **Unclear if the proposed autoencoder model is an improvement** The reported results do not include error bars or statistical analysis making it difficult to assess if the proposed autoencoder model is indeed better. Error bars should be available for Fig. 5. Also, the authors did not state why the simpler autoencoder models outperform their autoencoder model on various genes in Fig. 5 - this should at least be discussed.

3.  **Missing comparisons.** The authors state that point process models (line 76) can address invariance issues in MEA data, but they do not compare to these models in their analysis. Furthermore, self-attention is known to possess invariance properties [1] and it would be useful to compare these models too.

[1] Lee, J., Lee, Y., Kim, J., Kosiorek, A., Choi, S. and Teh, Y.W., 2019, May. Set transformer: A framework for attention-based permutation-invariant neural networks. In International conference on machine learning (pp. 3744-3753). PMLR.

**Questions:**

### Questions:
- In the abstract, the authors state the deep learning MEA analysis has mostly been focused on in vivo recordings and not in vitro recordings. Why can't the same methods be applied to in vitro recordings?
- What are excitability phenotypes (line 32)?
- Is the loss of resolution temporal or spatial (line 45)?
- Why would the compression of neural measures impact the quality of phenotype and disease models (line 48)?
- Why is it important for MEA recordings to be invariant to changes in the orientations of the electrodes (line 71)?
- What is "the disease" in line 391?
- What are the treatment similarities (line 433)? The authors state that the Roll-AE embeddings can be used to characterize treatment similarities, but I could not find them in the main text.
- Is the electrophysiological dataset publicly available?
- Is the source code publicly available?

### Additional feedback to improve your paper:
- I would reference the Figures in the main text in the order they appear, otherwise, the reader has to jump back and forth between the pages (minor point).
- The introduction seems very long. Although I appreciate the various backgrounds, it makes it harder to decipher what the paper is trying to address (minor point).
- Fig. 4a y-label might be wrong. Missing metric used for the scores (e.g. MSE or %).
- Unclear what Fig. 4b labels are? Also, I would include a colorbar to denote what the colors in the matrices mean.
- Fig. 5 and 6 font size is really small and it is hard to read. I would increase the font size.
- Consider using different colours in the bar plots in Fig. 5 as it is hard to see the light-coloured bars against the white background.

---

### Official Review · Reviewer_woME · 2024-10-21

**Soundness:** 2
**Presentation:** 2
**Contribution:** 2
**Rating:** 3
**Confidence:** 4

**Summary:**

This paper introduces a representation learning architecture that enforces shift-invariance to time-series spike data during training of a standard autoencoder architecture. The goal of the model is to learn reliable embeddings for different temporal rotations of the input, by performing a set-to-set mapping, through a linear assignment loss. Experiments are performed on microelectrode array data to demonstrate its effectiveness in extracting spatiotemporally invariant features from electrophysiological recordings.

**Strengths:**

- Empirically the proposed method appears more effective than vanilla data augmentation based AEs to enforce shift invariance.

**Weaknesses:**

- Unique contribution of the work is not very clear. It appears like the methodological ML contribution is limited, and is essentially similar to existing 2D/3D orientation-invariance approaches being adapted to spike-train shift invariance.

- Presentation and writing could also be improved. Figures with results are not very clear to read with very small text font sizes. Literature coverage is shallow.

**Questions:**

- Did the authors investigate visualizations (via UMAP etc.) of the latent representations obtained for different temporal shifts of the input?

- Can the authors elaborate why is the proposed Roll-AE architecture declared as a "foundational model"?

- Given the computational overhead of Eq (1), training load of such a model should be quantitatively elaborated in a table with comparisons.

- There are currently no comparisons to any existing invariant representation learning methods designed for autoencoding models. Furthermore, invariant representation learning from neurophysiological data [1-3] has also been extensively studied based on various loss- or model- regularization techniques in different contexts. This work lacks such discussions or any comparisons of its methodology with existing methods.

[1] "Learning time-invariant representations for individual neurons from population dynamics"

[2] "Capturing cross-session neural population variability through self-supervised identification of consistent neuron ensembles"

[3] "Deep site-invariant neural decoding from local field potentials"

---

### Official Review · Reviewer_FCu8 · 2024-11-03

**Soundness:** 3
**Presentation:** 1
**Contribution:** 2
**Rating:** 3
**Confidence:** 3

**Summary:**

The study introduces Roll-AE, an autoencoder designed to extract consistent features from neuronal activity recordings, particularly from stem cell-derived neuronal cultures. Traditional MEA data analysis is often limited by information loss, dependence on arbitrary settings, and challenges with missing data. Roll-AE overcomes these issues by incorporating invariance to temporal shifts and spatial electrode permutations. It processes sets of cyclic permutations of spike trains and uses an aggregation function to ensure consistent embeddings. The authors show that Roll-AE outperforms standard autoencoders in mimicking various neuronal firing patterns on synthetic datasets. Additionally, Roll-AE effectively captures meaningful phenotypes in neuronal cultures treated with siRNA, excelling in classification tasks, treatment clustering, and predicting traditional neural metrics.

**Strengths:**

The paper introduces Roll-AE, a novel autoencoder designed to manage spatiotemporal invariance in MEA recordings, moving beyond traditional methods that rely on hand-crafted features or standard autoencoders. It innovatively addresses temporal shifts and spatial permutations due to electrode symmetries, which are common challenges in in vitro MEA data analysis. The methodology is well-executed, with comprehensive experiments on synthetic and biological data providing strong evidence for Roll-AE's effectiveness. Roll-AE's ability to capture complex neuronal firing patterns and biologically relevant phenotypes without predefined metrics holds potential for advancing research in disease modeling, drug discovery, and understanding neuronal networks.

**Weaknesses:**

Processing all cyclic and electrode permutations for shift and spatial invariance can be computationally heavy, especially with large datasets or longer spike trains. The authors use stochastic shift-invariance to ease this but could further explore efficiency versus performance trade-offs. The paper lacks a detailed discussion on Roll-AE's limitations, such as potential information loss from permutation aggregation and sensitivity to specific patterns. It would also be helpful to see scenarios where Roll-AE might not excel. While Roll-AE could be applied to other spatiotemporal invariant data, there's no evidence or discussion on adapting it to other contexts like biomedical signals. Additionally, the impact of hyperparameters, such as the shift-sampling rate, on performance and computation could be more thoroughly examined to understand the model's robustness.

**Questions:**

Can the authors explain how Roll-AE's computational efficiency scales with larger datasets and suggest additional strategies to reduce computational load without losing performance? Have they tested different binning sizes to better capture detailed neural metrics? How could Roll-AE be adapted for other time-series data like ECG or sensor data? How does Roll-AE handle noise and missing data in real-world MEA recordings?  can the authors provide more insights into the interpretability of Roll-AE's embeddings, such as whether specific dimensions correspond to particular neuronal behaviors?

---

### Official Review · Reviewer_CC5Q · 2024-11-04

**Soundness:** 3
**Presentation:** 3
**Contribution:** 2
**Rating:** 5
**Confidence:** 3

**Summary:**

The authors introduce Roll-AE, an autoencoder designed for micro-electrode array data, which uses a set-theoretical inspired loss to obtain shift invariance. Latent embeddings of this model can be used to predict various metrics with better accuracy than latent embeddings of an autoencoder trained with a standard MSE loss.

**Strengths:**

1. The proposed architecture outperforms relevant baselines on the used benchmark.
2. The paper is well written and easy to understand.
3. A stochastic variant of the loss is introduced, which lowers the potentially prohibitive cost of the set-theory inspired loss.

**Weaknesses:**

1. All the quantitive evaluation is done “indirectly”, based on a decoder trained on latent embeddings of the different architectures.

	1.1. How do the models compare in terms of reconstruction loss? It might also be nice to show example reconstructions for the different architectures in the supplementary.

	1.2. It would be useful to include the statistics of memory and computation time (like Fig. 8) for the baseline architectures.
2. The baselines were not used in the Treatment Clustering and in the Neural Metrics Credentialing sections. It seems like they could also be used for each (but maybe with lower performance).
3. In Fig. 5 means are reported, adding error bars might be sensible.

**Questions:**

1. What makes this model a foundation model? I was of the opinion that this term is generally reserved for extremely large models trained on many datasets.
2. Using MLPs seems very costly in number of parameters — have the authors considered using e.g., CNNs with temporal convolutions?

---

### Meta-Review · Area_Chair_fG3b · 2024-12-21

**Metareview:**

This paper describes a novel auto-encoder that seeks to extract spatiotemporally invariant features using set-theoretic inspired loss function, with application to micro-electrode array recordings. The reviewers praised the paper's clarity, novelty, and the method's performance improvement over standard methods.  Unfortunately, however, they raised multiple critical issues, including the quantitative evaluation metrics, significance of the methodological advance, and relevance to other problem domains.  I regret that the paper in its current form cannot be accepted to this year's ICLR, but I wish the authors the best of luck in revising it for publication elsewhere.

**Additional Comments On Reviewer Discussion:**

The authors did not write a rebuttal, so there was no discussion during the rebuttal period.

---

### Decision · Program_Chairs · 2025-01-22

Reject